# Exploring the Antimicrobial Resistance Profile of *Salmonella typhi* and Its Clinical Burden

**DOI:** 10.3390/antibiotics13080765

**Published:** 2024-08-14

**Authors:** Muhammad Asghar, Taj Ali Khan, Marie Nancy Séraphin, Lena F. Schimke, Otavio Cabral-Marques, Ihtisham Ul Haq, Zia-ur-Rehman Farooqi, Susana Campino, Ihsan Ullah, Taane G. Clark

**Affiliations:** 1Institute of Pathology and Diagnostic Medicine, Khyber Medical University, Peshawar 25120, Pakistan; drazghar@gmail.com (M.A.); tajali.khan@medicine.ufl.edu (T.A.K.); 2Department of Pathology, Khyber Teaching Hospital, Khyber Medical College, Peshawar 25120, Pakistan; 3Emerging Pathogens Institute, University of Florida, Gainesville, FL 32611, USA; nseraphin@ufl.edu; 4Department of Medicine, Division of Infectious Diseases and Global Medicine, College of Medicine, University of Florida, Gainesville, FL 32610, USA; 5Department of Immunology, Institute of Biomedical Sciences, University of São Paulo, São Paulo 05508-220, Brazil; lenaschimke@hotmail.com (L.F.S.); otaviocmarques@gmail.com (O.C.-M.); 6Department of Medicine, Division of Molecular Medicine, University of São Paulo School of Medicine, São Paulo 05403-010, Brazil; 7Laboratory of Medical Investigation 29, University of São Paulo School of Medicine, São Paulo 05403-010, Brazil; 8Network of Immunity in Infection, Malignancy, Autoimmunity (NIIMA), Universal Scientific Education and Research Network (USERN), São Paulo 05508-000, Brazil; 9Department of Physical Chemistry and Technology of Polymers, Silesian University of Technology, M. Strzody 9, 44-100 Gliwice, Poland; ihaq@polsl.pl; 10Joint Doctoral School, Silesian University of Technology, M. Strzody 9, 44-100 Gliwice, Poland; 11Postgraduate Program in Technological Innovation, Federal University of Minas Gerais, Belo Horizonte 31270-901, Brazil; 12Khyber Medical University, Institute of Health Sciences, Swabi, Pakistan; farooqiscientist@gmail.com; 13Faculty of Infectious and Tropical Diseases, London School of Hygiene and Tropical Medicine, London WC1E 7HT, UK; susana.campino@lshtm.ac.uk; 14Faculty of Epidemiology and Population Health, London School of Hygiene and Tropical Medicine, London WC1E 7HT, UK

**Keywords:** *Salmonella typhi*, clinical paradigm, antibiogram, drug resistance, typhoid fever

## Abstract

**Background**: Typhoid fever caused by *Salmonella enterica* serovar Typhi *(S. typhi)* continues to pose a significant risk to public health in developing countries, including Pakistan. This study investigated the epidemiological factors linked to suspected and confirmed *S. typhi* infections in Peshawar’s hospital population. **Methodology**: A total of 5735 blood samples of patients with suspected enteric fever were collected from September 2022 to November 2023. *S. typhi* infection was confirmed using microbiological culture of blood samples, biochemical-based tests, and DNA-sequencing methods. Drug sensitivity testing on cultures was conducted as per the CLSI guidelines. Chi-square tests were used to analyze the clinical and epidemiologic characteristics of 5735 samples stratified by *S. typhi* infection status, and risk factors were assessed by applying logistic regression models to estimate odds ratios (ORs). **Results**: The number of confirmed typhoid fever cases in this hospital-based study population was 691 (/5735, 12.0%), more prevalent in males (447/3235 13.8%) and children (0–11 years) (429/2747, 15.6%). Compared to children, the risk of *S. typhi* infection was lower in adolescence (adjusted OR = 0.52; 95% CI: 0.42–0.66), adulthood (19–59 years; aOR = 0.30; 95% CI: 0.25–0.38), and older adulthood (aOR = 0.08; 95% CI: 0.04–0.18) (*p* < 0.001). Compared to males, the risk of *S. typhi* infection was lower in females (aOR = 0.67; 95% CI = 0.56–0.80; *p* = 0.002). Living in a rural residence (compared to urban) was associated with a higher risk of infection (aOR = 1.38; 95% CI: 1.16–1.63; *p* = 0.001), while access to a groundwater source (compared to municipal water supply) led to a lower risk (aOR = 0.56; 95% CI: 0.43–0.73; *p* = 0.002). Vaccination demonstrated a robust protective effect (aOR = 0.069; 95% CI = 0.04–0.11, *p* = 0.002). For those with typhoid infections, clinical biomarker analysis revealed the presence of leucopenia (65/691, 9.4%), thrombocytopenia (130/691, 18.8%), and elevated alanine aminotransferase (ALT) (402/691, 58.2%) and C-reactive protein (CRP) (690/691, 99.9%) levels. Worryingly, among the positive *S. typhi* isolates, there was a high prevalence of drug resistance (653/691), including multidrug-resistant (MDR 82/691, 11.9%) and extensively drug-resistant types (XDR, 571/691, 82.6%). **Conclusions**: This study highlights the importance of age, sex, locality, water source, and vaccination status in shaping the epidemiological landscape of *S. typhi* in the Peshawar district. It implies that expanding vaccination coverage to the broader population of Khyber Pakhtunkhwa province, particularly in the district of Peshawar, would be beneficial.

## 1. Introduction

*Salmonella enterica* serovar typhi *(S. typhi*) is a Gram-negative, rod-shaped, non-spore-forming bacterium that causes enteric fever and systemic infection, and it is common in geographical regions with poor sanitation infrastructure [1,2]. A total of 11–18 million cases of enteric fever, with approximately 1% mortality, have been reported worldwide each year. The disease burden is high in Africa and Asia due to underdeveloped sanitation infrastructures [3]. The *S. typhi* bacterium is transmitted to humans through the fecal–oral route and causes typhoid fever [4,5]. After entry into the human body, *S. typhi* mainly colonizes the ileum, liver, spleen, bone marrow, gallbladder, and blood, where it can cause bacteremia and septicemia [6]. Typhoid infection is characterized by high fever, fatigue, abdominal pain, and diarrhea [7]. The complications of typhoid fever include leucopenia and thrombocytopenia accompanied by elevated C-reactive protein (CRP) and alanine aminotransferase (ALT) levels [8,9]. Contaminated water and poor socioeconomic conditions contribute to the spread of enteric fever [10]. Pakistan was the first country to incorporate the World Health Organization (WHO)-recommended typhoid conjugate vaccine (TCV) into its regular immunization program in 2019 to combat typhoid fever [11,12]. Vaccination appears to be highly effective at reducing the risk of infection [13].

Until the 1970s, first-line treatments, including beta-lactams such as ampicillin, chloramphenicol, and cotrimoxazole (trimethoprim–sulfamethoxazole), were still effective, but now are unavailable due to widespread drug resistance (DR). The first-, second-, and third-line antibiotics are not effective against *S. typhi* due to the emergence of multidrug-resistant (MDR) and extensively drug-resistant (XDR) strains [14]. XDR *S. typhi* strains are highly prevalent, which could lead to a situation like that of the 1940s when typhoid fever was an incurable illness associated with a high mortality rate [15].

An outbreak of XDR *S. typhi* in Pakistan was first noted in Hyderabad, Sindh, in 2016 [16,17], where the infections also spread to Punjab, the largest province [1,18]. The largest city in Pakistan, Karachi, was severely impacted when this upsurge expanded to other regions of the province [12]. A total of 14,360 XDR cases of typhoid fever (XDR-TF) were reported in Karachi from January 2017 to June 2021, and 5741 XDR-TF cases were recorded in all Sindh provinces from November 2016 to June 2021 [11]. The poor health system in Pakistan plays a significant role in the emergence of XDR strains of *S. typhi* [19].

This study comprehensively examines the prevalence, antimicrobial resistance, and clinical epidemiology of *S. typhi* infections in Peshawar, Khyber Pakhtunkhwa, Pakistan. Specifically, the paper aims to assess the incidence rates of *S. typhi* across different demographic groups and geographic locations, analyze the antimicrobial resistance patterns with a focus on MDR and XDR strains, and identify common clinical markers such as leucopenia, thrombocytopenia, elevated ALT, and CRP levels. Additionally, the study investigates the impact of sociodemographic factors like socioeconomic conditions, water sources, and rural versus urban residency on the spread of typhoid fever. It evaluates the effectiveness of the typhoid conjugate vaccine in reducing infection rates. The findings are intended to guide public health strategies and improve the region’s clinical management of typhoid fever.

## 2. Results

### 2.1. Study Population

A total of 5735 enteric fever patients with suspected *S. typhi* infections from hospital inpatient and outpatient clinics were analyzed. The majority of typhoid patients, 623 (90.1) individuals, had a history of three or more days of fever, and 68 (9.9) had travel history to a pandemic area.

There was a predominance of patients that were male (3235/5735, 56.4%), children (0–11 years, 2747/5735, 47.9%), from the Peshawar district (5033/5735, 87.8%; vs. Hangu 4.6%, Mardan 3.0%, other districts 4.6% (Table 1) and rural areas (3163/5735, 55.2%)) (Table 1). A confirmation of the presence of *Salmonella typhi* in the water source was conducted by screening some of the drinking water samples by the Public Health Laboratory and obtaining information from the patient/guardian. Most of the suspected cases relied on the provision of a municipality-based water source (4771/5735, 83.2%), while a minority were vaccinated with TCV (928/5735, 16.2%) (Table 1). The number of confirmed *S. typhi* infection cases in the hospital-based study population was 691 (/5735, 12.0%), with the majority being male (447/691, 64.7%), children (429/691, 62.1%), from the Peshawar district (609/691, 88.1%; vs. Hangu 4.6%, Mardan 4.1%, other districts 3.2% (Table 1) and rural areas (431/691, 64.2%)), as well as drinking municipality-sourced water (617/691, 89.3%), and being TCV unvaccinated (673/691, 97.4%) (all *p* < 0.02) (Table 1).

Among the hospital study population, the majority (5670/5735, 98.9%) had an average white blood cell count, while 65 (1.1%)—all *S. typhi*-positive—had abnormal counts (leucopenia) (Table 1).

A total of 5605 (/5735, 97.7%) patients had average platelet counts, while 103 (1.8%) and 27 (0.5%) were mild and moderate cases, respectively, of thrombocytopenia. All thrombocytopenia cases were typhoid-positive patients (130/130). In the study population, 402 (7.0%) patients had increased ALT levels, while 690 (12.0%) had high CRP levels. Once again, all patients with elevated ALT or CRP levels were typhoid-confirmed positive cases (Table 1).

### 2.2. S. typhi-Confirmed Cases

The number of *S. typhi* infection cases in the hospital-sourced study population was 691 (/5735, 12.0%). The prevalence of *S. typhi* infection (12.0%) varied by epidemiological factors, including differences by sex (male 447/3235 (13.8%) vs. female 244/2500 (9.8%); *p* < 0.001) (Table 2). In this cohort, the prevalence varied significantly with age, with children having the highest burden (429/2747, 15.6%), compared to adolescents (110/896, 12.3%) and all adults (152/2092, 7.3%), with the subset of older adults (>60 years) having the lowest rates (6/262, 2.3%) (*p* < 0.001). This decreasing trend with age suggests a low level of immunity against *S. typhi* in younger age groups, potentially because of low vaccine coverage in recent years. Notwithstanding the potential uneven coverage of some districts and dominance of sampling from patients from Peshawar, there are some geographical similarities in the prevalence of *S. typhi* infection (Peshawar 609/5033 (12.1%), Hangu 32/264 (12.1%), Malakand 14/150 (9.3%), Mardan 28/174 (16.1%), other districts 8/114 (7.0%)). However, the prevalence of *S. typhi* infections was higher in hospital patients from rural (431/3163, 13.6%) compared to (peri-)urban areas (260/2572, 10.1%) (*p* < 0.001), as well as those living in communities using municipal water (617/4771 (12.9%) vs. groundwater supply 74/964 (7.7%)) (*p* < 0.001) (Table 2). As expected, the prevalence of *S. typhi* infections was higher in non-vaccinated (673/4807, 14.0%) compared to vaccinated patients (18/928, 1.9%) (*p* < 0.001) (Table 2).

### 2.3. Logistic Regression Model Using Typhoid Cases (Negative/Positive) as the Dependent Variable and Its Association with Other Covariates

Multivariate logistic regression analysis (Model 2) was conducted in Table 2 to examine the relationship between typhoid cases (both negative and positive) in the form of adjusted odds ratio (AOR) and various covariates in the Peshawar district of Khyber Pakhtunkhwa, and several key findings emerged. Our results showed that the age groups of 12–18 (adolescence), 19–59 (adulthood), and >60 (old age) were all significantly associated with reduced odds of typhoid cases. The odds ratios for these age groups were 0.52 (AOR) [95% CI 0.419–0.664, *p*-value 0.001], 0.30 (AOR) [0.252–0.378, *p*-value 0.001], and 0.08 (AOR) [95% CI 0.035–0.181, *p*-value 0.001], respectively, indicating a decreasing likelihood of typhoid cases with increasing age. Similarly, being female was associated with lower odds of typhoid cases, with an odds ratio of 0.67 (AOR) [95% CI 0.56–0.80, *p*-value 0.002]. Living in rural areas was associated with higher odds of typhoid cases, with 1.38 (AOR) [95% CI 1.16–1.63, *p*-value 0.001]. In comparison, peri-urban residence tended to lower odds, although this was not statistically significant (odds ratio 0.26, *p*-value 0.007). Using groundwater as a source was associated with lower odds of typhoid cases (odds ratio 0.56, *p*-value 0.002) compared to the municipality as a water source. Furthermore, those having a vaccination status of “Yes” were significantly associated with reduced odds of typhoid cases, with an AOR of 0.069 [95% CI 0.04–0.11, *p*-value 0.001].

### 2.4. Antibiogram of S. typhi Isolates

*S. typhi* bacteria were successfully isolated from the 691 patient samples for antibiogram testing, as per the CLSI guidelines (2023). The results of the antibiogram revealed that *S. typhi* isolates had the highest resistance to ampicillin (96.8%) and chloramphenicol (94.0%), followed by cotrimoxazole (93.0%), ciprofloxacin (91.7%), cefixime (90.5%), and ceftriaxone (88.5%) (Figure 1). Moreover, antibiotics such as azithromycin and meropenem were effective (Figure 1). Carbapenem (meropenem) was the drug of choice for all the extensively drug-resistant *S. typhi* patients. Using the *S. typhi* antibiogram data, 571 (/691, 82.6%) were classified as XDR, 82 (11.9%) were MDR, and only 38 (5.5%) were non-resistant across the panel of drugs. The high prevalence of XDR typhoid is a significant concern, as it represents a complex treatment challenge for clinicians.

## 3. Discussion

### 3.1. Prevalence and Geographic Variations

Typhoid fever stands as a significant infection in South Asia, particularly in Pakistan, where its prevalence remains notably high [20]. The incidence is estimated to be >100 per 100,000 people. Approximately seven million people are affected each year in South Asia, with approximately 75,000 deaths [21]. Among countries in South Asia, Pakistan has the highest estimated incidence rate of typhoid fever (493.5 per 100,000 persons/year) [22]. The reasons behind the elevated occurrence of typhoid fever in Pakistan are multifaceted [23]. The fluctuation in the prevalence of typhoid fever varies throughout the year, necessitating further investigation into these fluctuations. Conversely, some districts, such as Kohat, N. Waziristan, and Mohmand, reported no positive cases. These variations emphasize the need for targeted interventions tailored to specific geographical areas. The current study reported a prevalence of 12.0% among *S. typhi* isolates, with 64.7% of cases in males and 35.3% in females. The high prevalence of typhoid cases (62.1%) was observed in the childhood age group (1–11 years), followed by the adulthood age group (19–59 years) at 21.1%. The study conducted in Karachi involved 1175 tests, with 207 yielding positive results. Among these positive cases, 20.58% were males, while 14.33% were females, consistent with our findings. A similar survey in Quetta reported a 14.63% positivity rate, with a higher prevalence in the 11–20 age group [24,25,26]. Notably, our data revealed a greater prevalence of typhoid in males compared to females, particularly in the age group of 1 to 15 years. This increased prevalence can be attributed to school-going children and those with weaker immune systems, aligning with the findings of [27] and a study in Bangladesh, both highlighting that typhoid fever is most prevalent among school-going children [28]. The greater incidence of typhoid cases in rural settings can be attributed to the lower quality of sanitation and other facilities in these areas, compared to urban and peri-urban settings. In rural settings, there may be challenges related to inadequate sanitation infrastructure, limited access to clean water, and a greater prevalence of environmental factors conducive to the spread of waterborne diseases. These conditions can contribute to an increased risk of typhoid transmission. Conversely, urban areas tend to have better-quality water and sanitation facilities, which can significantly reduce the likelihood of typhoid outbreaks. The relatively low number of cases in peri-urban settings may be due to a combination of factors, including intermediate sanitation and water quality levels, which fall between rural and urban standards. Urban areas are often characterized by factors such as overcrowding and limited access to clean water and sanitation, which contribute to the spread of typhoid fever. Urban slums are known for having higher incidence rates of typhoid due to these challenging living conditions [29].

### 3.2. Vaccination Status, Socioeconomic Factors, Antibiotic Resistance, and Clinical Data

Most of the suspected patients in the present study were non-vaccinated (83.8%) while 16.2% were vaccinated. The high numbers of typhoid cases were recorded in rural areas (55.2%) compared to urban areas (44.0%), with most patients being non-vaccinated. Additionally, most typhoid cases were observed among communities using municipal water (83.2%) as compared to groundwater (16.8%). Our study results showed statistically significant associations with typhoid fever prevalence and area of residence. Hygienic standards, substandard water provision, and prevailing socioeconomic circumstances are contributing factors for enteric fever. *S. typhi* primarily resides in polluted water and food sources, with individuals contracting typhoid fever through the consumption of these contaminated substances [10]. Vaccination against typhoid appears to be highly effective in reducing the risk of infection, and individuals residing in rural areas are more susceptible to the disease. These findings can guide public health strategies, emphasizing the importance of vaccination campaigns, particularly in rural areas, to mitigate the spread of typhoid fever [13]. Our data showed a higher prevalence in rural areas. Furthermore, typhoid fever was more prevalent (15.30%) in Quetta’s rural area compared to its urban area (13.45%) [24]. Several studies conducted in the region of Khyber Pakhtunkhwa (KP) showed a higher prevalence of typhoid and its resistance patterns [26,30,31]. In our study, *S. typhi* isolates showed the highest resistance against ampicillin (94.0%), followed by cotrimoxazole (93.0%), chloramphenicol (94.0%), ceftriaxone (88.7%), cefixime (90.5%), levofloxacin (74.0%), and ciprofloxacin (97.0%). These findings are consistent with a study conducted in Peshawar [26]. Another study showed that *S. typhi* exhibited higher sensitivity to imipenem (100%) and azithromycin (95%), unlike in our study, where azithromycin showed only 2.0% sensitivity, while meropenem showed similar findings of 100% sensitivity [32]. Similarly, most of the isolates were found to be XDR (82.6%) and MDR (11.9%), aligning with the study of Ali et al. [32] which showed 20% multidrug-resistant cases, whereas 47% of strains were extensively drug-resistant. Another study revealed that the majority were XDR *S. typhi* (46.1%) and MDR *S. typhi* (24.5%) strains, similar to our findings [33]. Moreover, XDR *S. typhi* was reported to be resistant to several antibiotics such as ciprofloxacin, cotrimoxazole, ampicillin, ceftriaxone, and chloramphenicol [34], congruent with our study’s results.

Hematological and clinical changes, such as anemia, leucopenia, thrombocytopenia, and subclinical disseminated intravascular coagulation, are expected to occur in typhoid fever patients [35]. We report high percentages of typhoid patients who developed leucopenia (9.4%), thrombocytopenia (18.8%), elevated ALT (58.2%), and elevated CRP (99.9%) levels. Leucopenia is considered a common hematological finding in typhoid fever patients. In our study, leucopenia was observed in only 9.4% of the patients, which was consistent with the findings of others [8,9]. Thrombocytopenia was present in 18.8% of the patients, which was slightly more significant than that reported by other investigations (~10%) [9,36]. No patients had evidence of disseminated intravascular coagulopathy, an observation supported by an earlier study [37]. The incidence of liver enzyme elevation in typhoid fever has been reported across various case series (range 22–52%) [8,38,39]. In this study, we observed a much greater incidence of elevated ALT levels (58.2%). The present study showed a correlation between increased CRP levels and typhoid disease and *S. typhi*, positive culture, consistent with other studies [40,41].

The emergence of ciprofloxacin-resistant *S. typhi* strains was reported nationally and globally [42]. This critical issue of drug resistance underscores the urgency of prudent antibiotic use, rigorous monitoring of resistance patterns, and the implementation of comprehensive strategies to combat antibiotic resistance in the context of typhoid fever treatment. Addressing this challenge is essential to ensure the continued effectiveness of antibiotics in managing this infectious disease.

One of the study’s limitations is that individuals who claimed to have had vaccinations were not tested for typhoid immunity. We did not follow up with the patients, resulting in insufficient data on the outcomes of different treatments for MDR and XDR *S. typhi* infections. This lack of outcome data poses a risk of inadequate decision-making in both the treatment of current infections and the prevention of future cases.

## 4. Materials and Methods

### 4.1. Study Design

A cross-sectional study was conducted at the Institute of Pathology and Diagnostic Medicine, Khyber Medical University, and the Peshawar and Pathology Department of Khyber Teaching Hospital (KTH), Peshawar. The current study was approved by the Institutional Review and Ethical Board of Khyber Medical College (KMC)/Khyber Teaching Hospital (Ref. 750/ Department of Medical Education (DME)/KMC, 24 November 2023).

### 4.2. Inclusion and Exclusion Criteria

All those suspected patients with a history of fever for 3 or more days or a history of travel within the last 28 days before the onset of illness for persons to endemic settings, as per the WHO case definition, were included in the present study. Patients who did not fall within the WHO criteria were excluded.

### 4.3. Data Collection 

Blood specimens were collected from each study participant in sterile blood culture bottles and tubes facilitated by the phlebotomists and laboratory technologists using the case investigation and blood culture form of the WHO. All the physicians were asked to complete a case investigation and blood culture form for the patient/guardian by the research team to gather information on clinical features, duration of disease, basic demographics, geographic distribution, locality, water source, and vaccine status. The vaccination status of the patients was verified by obtaining information from the patient/guardian.

### 4.4. Sample Collection and Processing

A total of 5735 samples of patients with suspected enteric fever were collected on first arrival to the hospital before treatment from September 2022 to November 2023 at KTH, using a WHO-based proforma (Appendix A). A total of 10 mL whole-blood samples were collected and transferred into culture bottles of the Versa Trek (Waterloo, WI, USA) automated blood culture system and then loaded into the Versa Trek for aerobic incubation at 37 °C for five days to detect gas production.

### 4.5. Description of Bacterial Culture, Controls, Chemical Compositions of Media, and Antibiotics

Before use, quality testing of all the examined clinical isolates was carried out using the *S. typhimurium* ATCC 14,028 control strain. *S. typhi* ATCC 6539 was used as a quality control organism for sensitive control. *E. coli* ATCC 25922 was used for resistance control. All media used in this study for *S. typhi*, such as Blood Agar, MacConkey Agar, Salmonella Shigella Agar, Mueller Hinton Agar, and Blood Culture Media Bottle Versa TREK Automated Microbial Detection System (USA) were tested for quality control, sterility, physical appearance, and biochemical reactivity using the ATCC organisms like *Escherichia coli* ATCC 25922, *Staphylococcus aureus* ATCC 25923, *Pseudomonas aeruginosa* ATCC 27853, and *Shigella sonnei* ATCC 25931. Additionally, quality control for all tested antibiotics was performed before use, utilizing the ATCC strains, with breakpoints interpreted based on CLSI 2023 [43]. Antibiotics including chloramphenicol (30 µg), cefotaxime (30 µg), ciprofloxacin (5 µg), ampicillin (10 µg), cotrimoxazole (25 µg), azithromycin (15 µg), and meropenem (10 µg) were applied to MHA plates and incubated overnight at 37 °C. After incubation, inhibition zones around the antibiotic discs were measured and compared to CLSI 2023 standards.

### 4.6. Phenotypic and Genotypic Identification 

The morphology of the pure isolated colonies was analyzed, and the colonies were subjected to Gram staining. Typhoid bacteria were determined using routine microscopic methods. For identification, the bacterial cultures were subjected to different biochemical tests using Analytical Profile Index (API) 20E strips (Biomerieux, Craponne, France). Routine clinical laboratory parameters, including the white blood cell (WBC) count, platelet count, ALT, and CRP, were also determined using standard protocols. The 16S rRNA gene is a highly conserved DNA sequence in the bacterial genome. Therefore, isolates were confirmed as *S. typhi* via 16S rRNA gene sequencing from the cultures.

### 4.7. Antibiotic Susceptibility Testing (AST)

This was performed using the Kirby–Bauer disc diffusion technique, per the Clinical and Laboratory Standard Institute 2023 (CLSI-2023) guidelines. Bacterial overnight broth cultures with a turbidity of 0.5 McFarland (a turbidity standard) were inoculated onto Mueller–Hinton agar (MHA) using sterile cotton swabs. Antibiotic discs were applied to MHA plates and incubated overnight at 37 °C. Following overnight incubation, inhibition zones around the antibiotic discs were measured and compared with the CLSI 2023 standards.

### 4.8. Epidemiological and Statistical Analysis

The suspected and confirmed typhoid cases were analyzed by sex, age groups (childhood (0–11 years), adolescence (12–18 years), adulthood (19–59 years), older adults (60+ years)), geographic area (district), local water source, and TCV vaccination status. Chi-square tests were used to evaluate the samples’ clinical presentation and epidemiologic characteristics stratified by *S. typhi* infection status (infected (tested positive) vs. suspected but tested negative). The logistic regression models were fitted to assess associations between *S. typhi* infection status and epidemiological characteristics or risk factors, leading to odds ratios (ORs). A *p*-value of <0.05 was considered to indicate statistical significance. The analyses were conducted using IBM SPSS Statistics (v23.0.0) (SPSS, Inc., Chicago, IL, USA).

## 5. Conclusions

The results of this extensive analysis on typhoid fever prevalence and related factors provide valuable insights into the dynamics of this infectious disease within the population studied. This multifaceted analysis of typhoid fever prevalence in Khyber Pakhtunkhwa, Pakistan, underscores the importance of considering gender, time, age, and location when developing public health strategies. The data reveal notable disparities and variations, highlighting opportunities for targeted interventions and further research to better understand and mitigate the spread of typhoid fever in the region.

## Figures and Tables

**Figure 1 antibiotics-13-00765-f001:**
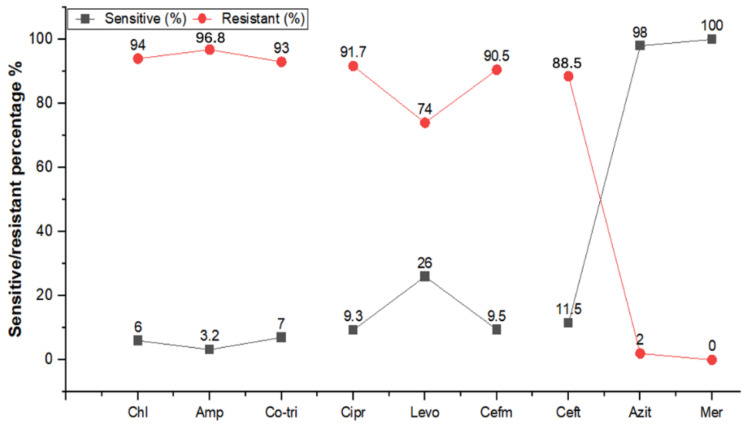
Antibiogram of confirmed *S. typhi* isolates (n = 691) from Peshawar, Khyber Pakhtunkhwa, Pakistan, from September 2022 to November 2023. Chl, chloramphenicol; Amp, ampicillin; Co-tri, cotrimoxazole; Cipr, ciprofloxacin; Levo, levofloxacin; Cefm, cefixime; Ceft, ceftriaxone; Azit, azithromycin; Mer, meropenem.

**Table 1 antibiotics-13-00765-t001:** Characteristics of the study population and correlation with *S. typhi* infection status.

Variable	Categories	Population (%)	*S. typhi* Infection Status		*p*-Value
			Negative N (%)	Positive N (%)	
Gender	Male	3235 (56.4)	2788 (55.3)	447 (64.7)	0.001
	Female	2500 (43.6)	2256 (44.7)	244 (35.3)	
Age groups	Childhood (0–11 years)	2747 (47.9)	2318 (46.0)	429 (62.1)	0.001
	Adolescence (12–18 years)	896 (15.6)	786 (15.6)	110 (15.9)	
	Adulthood (19–59 years)	1830 (31.9)	1684 (33.4)	146 (21.1)	
	Older Adults (60+ years)	262 (4.6)	256 (5.1)	6 (0.9)	
Geographic distribution	Peshawar	5033 (87.8)	4424 (87.7)	609 (88.1)	0.014
	Hangu	264 (4.6)	232 (4.6)	32 (4.6)	
	Malakand	150 (2.6)	136 (2.7)	14 (2.0)	
	Mardan	174 (3.0)	146 (2.9)	28 (4.1)	
	Bannu	34 (0.6)	28 (0.6)	6 (0.9)	
	Kohat	80 (1.4)	78 (1.5)	2 (0.3)	
Locality	Urban	2522 (44.0)	2264 (44.9)	258 (37.3)	0.002
	Rural	3163 (55.2)	2732 (54.2)	431 (62.4)	
	Peri-Urban	50 (0.9%)	48 (1.0)	2 (0.3)	
Water Source	Municipality	4771 (83.2)	4154 (82.4)	617 (89.3)	0.001
	Ground Water	964 (16.8)	890 (17.6)	74 (10.7)	
TCV Vaccination Status	No	4807 (83.2)	4134 (82.0)	673 (97.4)	0.002
	Yes	928 (16.2)	910 (18.0)	18 (2.6)	
History of Fever	Three or >3 days	4128 (72.0)	3505 (69.5)	623 (90.1)	0.002
	Recent Travel	1607 (28.0)	1539 (30.5)	68 (9.9)	
Leucopenia	Normal cases (4000–11,000/cm)	5670 (98.9)	5044 (100)	626 (90.6)	0.002
	Abnormal cases (<4000/cm)	65 (1.1)	0 (0)	65 (9.4)	
Thrombocytopenia	Normal cases (150,000–450,000/cm)	5605 (97.7)	5044 (100)	561 (81.2)	0.001
	Mild (50,000–149,000/cm)	103 (1.8)	0 (0.0)	103 (14.9)	
	Moderate (30,000–50,000/cm)	27 (0.50)	0 (0.0)	27 (3.9)	
ALT/SGPT	Normal cases (<45 U/L)	5333 (93.0)	5044 (100)	289 (41.8)	0.001
	Abnormal cases (>45 U/L)	402 (7.0)	0 (0)	402 (58.2)	
CRP	Normal cases (up to 5 mg/dl)	5045 (88.0)	5044 (100)	1 (0.1)	0.002
	Abnormal cases (>5 mg/dl)	690 (12.0)	0 (0)	690 (99.9)	

ALT, alanine transaminase; SGPT, serum glutamic pyruvic transaminase; CRP, C-reactive protein; TCV, typhoid conjugate vaccine.

**Table 2 antibiotics-13-00765-t002:** Factors associated with a confirmed *S. typhi* infection in this hospital-based cohort (n = 5735).

Characteristics	Univariate Analysis			Multivariate Analysis		
	OR	95%CI	*p*-Value	OR	95%CI	*p*-Value
Age group (in years)						
12–18 (adolescence) vs. child *	0.75	0.60–0.94	0.001	0.52	0.419–0.664	0.001
19–59 (adulthood) vs. child *	0.46	0.38–0.57	0.001	0.30	0.252–0.378	0.001
>60 (old age) vs. child *	0.12	0.05–0.28	0.003	0.08	0.035–0.181	0.001
Gender						
Female vs. male	0.67	0.57–0.79	0.001	0.67	0.56–0.80	0.002
Locality						
Rural vs. urban	1.38	1.17–1.63	0.001	1.38	1.16–1.63	0.001
Peri-urban vs. urban	0.36	0.08–1.51	0.366	0.26	0.06–1.11	0.007
Water source						
Groundwater vs. municipal	0.56	0.43–0.72	0.001	0.56	0.43–0.73	0.002
TCV vaccination status						
Yes vs. no	0.12	0.07–0.19	0.001	0.07	0.04–0.11	0.001

* 0–11 years (childhood); OR, odds ratio.

## Data Availability

Data available in the article and the Appendix A which includes the whole-genome analysis of salmonella isolates, WHO-based proforma used for the collection of blood samples, and WHO Classification of Typhoid fever cases by drug resistance status.

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
