# Peer review of "Exploring the Antimicrobial Resistance Profile of Salmonella typhi and Its Clinical Burden"

_antibiotics, 2024, doi:10.3390/antibiotics13080765_

Round 1

Reviewer 1 Report

Comments and Suggestions for Authors

The document "Exploring Antimicrobial Resistance Profile of Salmonella typhi and its Clinical Burden" proposes an interesting study on the spread of S. typhi in a hospital population in Peshawar. S. typhi represents a significant risk to public health in developing countries. The authors analysed a total of 5,735 blood samples from patients with suspected enteric fever. The data were collected through a comprehensive study of patients suspected of being infected with S. typhi. The methods were as clear as the results. The collected data provide a clear and comprehensive overview of the epidemiology of S. typhi in Peshawar, through the analysis of the correlation between the impact of the environment and vaccination on the spread of infection. The discussion allows you to analyze the issue from multiple angles. Overall, the document allows the implementation of public health strategies to regulate the irrational use of antibiotics and reduce antimicrobial resistance (AMR). An interesting contribution to public health

Author Response

The document "Exploring Antimicrobial Resistance Profile of Salmonella typhi and its Clinical Burden" proposes an interesting study on the spread of S. typhi in a hospital population in Peshawar. S. typhi represents a significant risk to public health in developing countries. The authors analysed a total of 5,735 blood samples from patients with suspected enteric fever. The data were collected through a comprehensive study of patients suspected of being infected with S. typhi. The methods were as clear as the results. The collected data provide a clear and comprehensive overview of the epidemiology of S. typhi in Peshawar, through the analysis of the correlation between the impact of the environment and vaccination on the spread of infection. The discussion allows you to analyze the issue from multiple angles. Overall, the document allows the implementation of public health strategies to regulate the irrational use of antibiotics and reduce antimicrobial resistance (AMR). An interesting contribution to public health

We are immensely grateful to reviewer for recognizing our work.

We 

Reviewer 2 Report

Comments and Suggestions for Authors

The manuscript should be considered for publication after some minor corrections/revisions are made.

First, I think there is a general correction to be made throughout the text.
Scientific papers are preferably written in impersonal language, but the majestic plural and the narrative style are also allowed.
Regardless of the style adopted, it is important to maintain coherence and consistency throughout the work (i.e. only one of the three styles should be adopted, which should be maintained throughout the text). That's not the case here. For example, in the abstract (lines 37-38; 53-57), materials and methods (lines 145-147) and results (lines 199-200) personal language is used while in most of the text the impersonal form is used (e.g. in lines 96-97 "....the present study aimed to investigate the prevalence ...."). I suggest uniformization.

Introduction
lines 72-73: The sentence "The provision of contaminated water with poor socioeconomic conditions contributes to enteric fever." is not correct, since socioeconomic conditions are characteristics that can be attributed to people or locations/regions but not water. Please rewrite.

Materials and Methods
line 99: the section "2. Materials and Methods", as well as the following section "3. Results" (line 152), should be relocated to the top of the following page.
line 126: “… further also …” is not correct. Please amend.

Results
line 157: a parenthesis should be added after "... 4.6%", thus changing "......4.6%; Figure 1) …" to "......4.6%) (Figure 1) …"
line 163: the same as the previous comment; it should be changed to "... districts 3.2%) (Figure 1) ..."
line 174: the information contained in Table 1 is not so vast as to prevent it from being fully located on the same page, thus allowing for an easier analysis of the results; therefore, I believe that there should be a repositioning in the text so that it is fully located on the same page.
line 197: the heading of Table 2 should be next to the table itself; please correct.

Discussion
line 285: please replace the comma after the references with a period (i.e. "..."). [8,9], Thrombocytopenia ..." to "... 8,9]. Thrombocytopenia ...")
lines 295-298: it is stated that: "Our study revealed that isolates of S. typhi exhibited the highest resistance to first-, second- and third-line antibiotics."; immediately after comes the sentence: "These drugs also proved effective against S. typhi in a study conducted in Peshawar [26]."; there seems to be a contradiction between the two
sentences; please alter.

Author Response

We sincerely appreciate the valuable feedback from the reviewer. These revisions have significantly improved the manuscript, strengthening the scientific arguments and clarifying the findings. Each modification has been highlighted in the revised manuscript for ease of identification.

Reviewer's comment

Response

First, I think there is a general correction to be made throughout the text. Scientific papers are preferably written in impersonal language, but the majestic plural and the narrative style are also allowed.

We appreciate the reviewer's positive feedback and have removed the personal language throughout the manuscript (Highlighted yellow).

Regardless of the style adopted, it is important to maintain coherence and consistency throughout the work (i.e. only one of the three styles should be adopted, which should be maintained throughout the text). That's not the case here. For example, in the abstract (lines 37-38; 53-57), materials and methods (lines 145-147) and results (lines 199-200) personal language is used while in most of the text the impersonal form is used (e.g. in lines 96-97 "....the present study aimed to investigate the prevalence ...."). I suggest uniformization.

We appreciate the reviewer's valuable comments. We have ensured the consistent use of impersonal language throughout the manuscript, both in the sections indicated by the reviewer and in other areas highlighted in yellow(Highlighted yellow).

Introduction

 lines 72-73: The sentence "The provision of contaminated water with poor socioeconomic conditions contributes to enteric fever." is not correct, since socioeconomic conditions are characteristics that can be attributed to people or locations/regions but not water. Please rewrite.

We are grateful to the reviewer for their insightful suggestions and have corrected this sentence accordingly (Highlighted yellow).

Materials and Methods line 99: the section "2. Materials and Methods", as well as the following section "3. Results" (line 152), should be relocated to the top of the following page. line 126: “… further also …” is not correct. Please amend

We are indebted to the reviewer for their positive comments and have relocated the materials, methods, and results to the top (Highlighted yellow).

Results line 157: a parenthesis should be added after "... 4.6%", thus changing "......4.6%; Figure 1) …" to "......4.6%) (Figure 1) …" line 163: the same as the previous comment; it should be changed to "... districts 3.2%) (Figure 1) ..."

We are obliged to the reviewer for their keen observations and have added parentheses in both places (Highlighted yellow).

line 174: the information contained in Table 1 is not so vast as to prevent it from being fully located on the same page, thus allowing for an easier analysis of the results; therefore, I believe that there should be a repositioning in the text so that it is fully located on the same page.

line 197: the heading of Table 2 should be next to the table itself; please correct.

We appreciate the reviewer's notes. We have placed entire Table 1 on a single page and positioned the heading of Table 2 next to it (Highlighted yellow).

Discussion line 285: please replace the comma after the references with a period (i.e. "..."). [8,9], Thrombocytopenia ..." to "... 8,9]. Thrombocytopenia ...")

We appreciate the reviewer's suggestions; we have removed the comma accordingly (Highlighted yellow)

lines 295-298: it is stated that: "Our study revealed that isolates of S. typhi exhibited the highest resistance to first-, second- and third-line antibiotics."; immediately after comes the sentence: "These drugs also proved effective against S. typhi in a study conducted in Peshawar [26]."; there seems to be a contradiction between the two sentences; please alter

We are pleased with the reviewer's positive feedback, and we have addressed the contradiction accordingly (Highlighted yellow)

Reviewer 3 Report

Comments and Suggestions for Authors

1.      The final part of the Introduction must be phrased better. The objectives of the study should be described with greater clarity. The authors should justify their study correctly. The authors must explain in detail the gaps in the literature that would be filled through publication of this manuscript. Moreover, please indicate what are the differences and the advantages of this manuscript in comparison with other relevant papers previously published.

Comment. To my belief, there is little novelty in this work, so the authors must try hard to explain why this needs to be published.

2.      Methodology. The authors do not describe at all the controls employed  in this study and that is a serious error. Please provide explanations for controls used: patients, bacteria type cultures, chemicals, antibiotics. All these must be included in a new sub-section to be added in the revised manuscript.

Comment. This is a serious omission, that if left unattended to, it will lead to immediate rejection.

3.      Methodology and results. Why did you not do a multivariable analysis of predictors? This should be rectified in the revised version.

Comment. A multivariable analysis will improve validity of the study.

4.      Discussion. Please subdivide in two sub-sections.

Comment. This will improve the readability of the manuscript.

5.      Conclusions. Conclusions should be brought in line with the findings of the study. Please rewrite from scratch.

Comment. The conclusions should not vary significantly from the findings of the study. Only a small extrapolation can be allowed.

Overall. Significant improvement after extensive revision and resubmission for possible acceptance; recommendation for acceptance is not guaranteed, unless real improvement will be seen.

Author Response

Reviewer's comment

Response

The final part of the Introduction must be phrased better. The objectives of the study should be described with greater clarity. The authors should justify their study correctly. The authors must explain in detail the gaps in the literature that would be filled through publication of this manuscript. Moreover, please indicate what are the differences and the advantages of this manuscript in comparison with other relevant papers previously published.

 We are grateful to the reviewer for their observations. We have enhanced the clarity and highlighted the importance of the objective section. Additionally, we have emphasized factors that have not been reported before (Highlighted yellow).

.      Methodology. The authors do not describe at all the controls employed  in this study and that is a serious error. Please provide explanations for controls used: patients, bacteria type cultures, chemicals, antibiotics. All these must be included in a new sub-section to be added in the revised manuscript.

We are deeply grateful to the reviewer for pointing out these significant shortcomings. In response, we have incorporated a new paragraph titled "Description of bacterial culture, controls, chemical compositions of media, and antibiotics," which provides comprehensive information on these aspects. Undoubtedly, adding such valuable details has improved our manuscript (Highlighted yellow).

Methodology and results. Why did you not do a multivariable analysis of predictors? This should be rectified in the revised version.

We appreciate the reviewer for their insightful comments. We would like to clarify that both univariate and multivariable analyses using logistic regression are presented in Table 2 of the manuscript. We apologize if this was unclear initially, and we have made further revisions to emphasize the presence and significance of these analyses more explicitly in the text (Highlighted yellow).

Discussion. Please subdivide in two sub-sections.

We deeply value the reviewer's valuable suggestions. As a result, we have divided the discussion into two subsections (Highlighted yellow).

Conclusions. Conclusions should be brought in line with the findings of the study. Please rewrite from scratch.

We are thankful to the reviewer for their significant observations. As a result, we have improved the conclusion to clearly present the findings of this study (Highlighted yellow).

Round 2

Reviewer 3 Report

Comments and Suggestions for Authors

The authors have not taken into account and have not clarified all the issues raised.

For example, the results of the multivariable analysis are badly presented and the Discussion is still in one single section (despite the authors mentioning that they sub-divided).

Author Response

Reviewer Comments

Response

The authors have not taken into account and have not clarified all the issues raised.

For example, the results of the multivariable analysis are badly presented and the Discussion is still in one single section (despite the authors mentioning that they sub-divided).

We are extremely thankful to reviewer comments and do apologies for these issues.

We revised the results of multivariable analysis (after table 4, Page 9). The discussion is sub divided. Apologies for the misunderstanding

Round 3

Reviewer 3 Report

Comments and Suggestions for Authors

.

Author Response

Response to Reviewer comments

Query: The authors provide an extensive amount of detail on the laboratory process for confirming the S. typhi infection. However, very little of the Methods section is devoted to how the demographic and clinical characteristic data were compiled and defined. For example, was the drinking water source based on patient/guardian self-report or was this ascertained by the investigators based on the patient's address?

Response: We are thankful to reviewer for this comment. The information of drinking water source was obtain from the patient/guardian.

Query: Similarly, was vaccination status self-reported or were these data obtained by an examination of vaccine registries?

Response: We appreciate the reviewer comment. The vaccination status of the patients was verified by getting information from the patient/guardian.

Query: Do the blood sample metrics (e.g., CRP, platelets, etc.) presented reflect what was collected when the patient first arrived at the hospital or could some of these reflect blood collected after treatment had already been initiated?

Response: We are thankful to reviewer. The blood samples for all investigations including CRP, Platelets, etc. were collected from suspected patients on first arrival to the hospital before treatment. The changes are highlighted in the materials and methods section. 

Query: One of my main recommendations to the authors would be consider removing some of the extra detail about the laboratory process in the Methods section and devote some space to describing the patient-level data.

Response: We thank reviewer for this comment. The changes have been made as per suggestion, reduced the laboratory process data, and removed the tables in the materials and methods section. 

Query: All studies have implicit assumptions and limitations, and this study is no exception. I would recommend the authors insert a paragraph in the Discussion describing the possible limitations and threats to validity.

Response: We appreciate this comment of reviewer, we have added a paragraph about limitations after discussion section

Query: Figure 1 appears to be simply re-stating what's already listed in Table 3, so I'm unclear on why it's needed.

Response: We appreciate this comment. The figure 1 is omitted accordingly.

Query: Were there any other covariates in the multivariable model and only those factors considered significant listed in Table 4 or does Table 4 reflect the entire model?

Response: Thank you for your query. Table 4 (new number table 2) reflects only those covariates that were found to be significant in the Chi-square analysis. In line with standard regression analysis practices, we included only covariates that showed a significant association with the outcome. Non-significant covariates were excluded in the model to avoid potential overfitting and to present a more parsimonious model. The final model presented includes the unadjusted odds ratios (Model 1) and the adjusted odds ratios (Model 2) for the significant covariates in table 4 (new number table 2).

Query: Lastly, since the inclusion was either fever of three or more days *or* recent travel, I would also recommend the authors include the proportion of these criteria (e.g., Table 3).

Response: We appreciate this suggestion of reviewer. Proportion of these criteria has been added to result section, Table 3 (new number table 1).